# An embeddable molecular code for Lewis X modification through interaction with fucosyltransferase 9

Taiki Saito[1,2], Hirokazu Yagi [1,3], Chu-Wei Kuo[4], Kay-Hooi Khoo [3,4] & Koichi Kato [1,2,3✉]

N-glycans are diversified by a panel of glycosyltransferases in the Golgi, which are supposed to modify various glycoproteins in promiscuous manners, resulting in unpredictable glycosylation profiles in general. In contrast, our previous study showed that fucosyltransferase 9 (FUT9) generates Lewis X glycotopes primarily on lysosome-associated membrane protein 1 (LAMP-1) in neural stem cells. Here, we demonstrate that a contiguous 29-amino acid sequence in the N-terminal domain of LAMP-1 is responsible for promotion of the FUT9-catalyzed Lewis X modification. Interestingly, Lewis X modification was induced on erythropoietin as a model glycoprotein both in vitro and in cells, just by attaching this sequence to its C-terminus. Based on these results, we conclude that the amino acid sequence from LAMP-1 functions as a "Lewis X code", which is deciphered by FUT9, and can be embedded into other glycoproteins to evoke a Lewis X modification, opening up new possibilities for protein engineering and cell engineering.

---

[1] Graduate School of Pharmaceutical Sciences, Nagoya City University, 3-1 Tanabe-dori, Mizuho-ku, Nagoya 467-8603, Japan. [2] Institute for Molecular Science, National Institutes of Natural Sciences, 5-1 Higashiyama, Myodaiji-cho, Okazaki 444-8787, Japan. [3] Exploratory Research Center on Life and Living Systems (ExCELLS), National Institutes of Natural Sciences, 5-1 Higashiyama, Myodaiji-cho, Okazaki 444-8787, Japan. [4] Institute of Biological Chemistry, Academia Sinica, 128, Academia Road Sec. 2, Nankang, Taipei 115, Taiwan. ✉email: kkatonmr@ims.ac.jp

N-glycosylation, one of the most extensively studied post-translational modifications, is responsible for fine tuning the stability and functionality of secretory and membrane proteins[1]. N-glycans are attached to nascent proteins in the endoplasmic reticulum and diversified by a panel of glycosyl-transferases located in the Golgi apparatus as exemplified by fucosyltransferases, galactosyltransferases, and sialyltransferases. Cellular N-glycosylation profiles are thus generally unpredictable but can be biased through up- and down-regulation of these glycosyltransferases, which are supposed to modify various gly-coproteins in promiscuous manners, rendering the consequent effects non-selective for the glycoproteins produced in the cell[2–5].

Switching of N-glycosylation profiles controls cell fate and behavior during development, differentiation, aging, and patho-genic transformation[6–8]. This is exemplified by differentiation of neural stem cells promoted through the disappearance of the Lewis X moiety (i.e., Gal-β(1,4)-[Fuc-α(1,3)]-GlcNAc) from the outer branches of N-glycans. In neural stem cells, Lewis X is generated by the action of fucosyltransferase 9 (FUT9) and maintains proliferation through the activation of Notch signaling[9]. We found that the Lewis X modification occurs pri-marily on the N-glycans of lysosomal associated membrane protein 1 (LAMP-1)[10]. This unexpected non-promiscuous fuco-sylation suggests the existence of an unknown molecular mechanism mediating the fucosyltransferase and its specific substrate protein.

In this study, we attempt to elucidate the mechanism based on the assumption that a certain code leading to an encounter with FUT9 is hidden in the LAMP-1 molecule. Through a series of mutational analyses, we identified a unique molecular code recognized by FUT9. Furthermore, we demonstrate that this molecular code is embeddable to other glycoproteins to evoke the Lewis X modification. The biological and biotechnological sig-nificance of these findings will be discussed.

## Results

**FUT9-dependent Lewis X modification.** Our previous study revealed that LAMP-1 is the major substrate of FUT9 in mouse neural stem cells[9,10]. To determine the generality of this finding, we examined the effects of FUT9 overexpression on the N-glycosylation profiles in Lewis X-negative cells. Lysates pre-pared from CHO-K1 cells overexpressing FUT9 were subjected to immunoblot analysis using the anti-Lewis X antibody, AK97, and anti-LAMP-1 antibody, indicating a single prominent band cor-responding to an apparent molecular mass of 135 kDa (Fig. 1a). Also, the lysates treated with PNGase F exhibited a LAMP-1-positive band only at position corresponding to a 40-kDa protein (Fig. 1b), consistent with what is expected from the amino acid sequence and also with our previous observation[10]. These results strongly suggest that FUT9 catalyzes Lewis X modification solely on LAMP-1 in CHO-K1 cells as in the case of observation in neural stem cells. To confirm this finding, lysates prepared from FUT9-overexpressing CHO-K1 cells were immunoprecipitated with AK97 and then subjected to immunoblot analysis using the anti-LAMP-1 antibody. We found that the Lewis X-containing immunoprecipitants exhibited a single band reactive with anti-LAMP-1 antibody at the position corresponding to a molecular mass of 135-kDa, confirming that Lewis X was exclusively expressed on LAMP-1 (Fig. 1c). Conversely, immunoprecipitates formed with the anti-LAMP-1 antibody yielded Lewis X-positive bands (Fig. 1d). The Lewis X-positive band was abolished by LAMP-1 knockdown in the CHO-K1 cells overexpressing FUT9 (Fig. 1e). Furthermore, FUT9 overexpression in HEK293T cells and COS7 cells also resulted in selective Lewis X modifications of the 135-kDa protein (Supplementary Fig. 1). All these data

indicate that FUT9 promotes LAMP-1-specific Lewis X mod-ifications, not only in neural stem cells, but also in various mammalian cell lines.

Furthermore, we found that recombinant LAMP-1 underwent a Lewis X modification in FUT9-overexpressing CHO-K1 cells (Supplementary Fig. 2). This recombinant LAMP-1 glycoprotein was subjected to site-specific N-glycosylation profiling. LAMP-1 contains 18 putative N-glycosylation sites, of which LC-MS/MS analysis followed by Byonic search and manual interpretation of the data identified ten N-glycosylation sites at Asn62, Asn76, Asn84, Asn103, Asn130, Asn165, Asn181, Asn249, Asn261, and Asn322 (Supplementary Data 1 and 2). All of these N-glycosylation sites, except Asn322, carried complex-type N-glycans with at least one fucose. Among them, Asn76, Asn84, Asn130, Asn181, Asn249, and Asn261 exhibited the FUT9-dependent increments of fucosylation as revealed by MS analysis (Fig. 2a and Supplementary Fig 3). Figure 2b shows a representative MS/MS spectrum with an oxonium ion at $m/z$ 512.198 (HexNAc$_1$Hex$_1$Fuc$_1$) suggesting a Lewis X-containing glycan modification on Asn76.

**Identification of a LAMP-1 segment evoking a FUT9-dependent Lewis X modification.** To identify the possible determinants of FUT9-dependent Lewis X modification, we constructed a series of LAMP-1 mutants and expressed them in CHO-K1 cells with or without FUT9 overexpression (Fig. 3a). LAMP-1 is composed of two homologous immunoglobulin domains followed by transmembrane regions (Fig. 3b)[11]. First, we assessed whether the transmembrane region is required for FUT9-dependent Lewis X modification. Although the LAMP-1 mutant lacking the transmembrane region was secreted into the medium, unlike the wild type, it also underwent Lewis X mod-ification (Fig. 3c). This indicates that the transmembrane region of LAMP-1 is dispensable for its selective encounter with FUT9, suggesting that the luminal region carries the critical determinant for Lewis X modification. To identify this determinant, we expressed the N- or C-domain alone in FUT9-overexpressing CHO-K1 cells and examined the resulting Lewis X modification. The results showed that FUT9-dependent Lewis X modification occurred only in the N-domain, but not the C-domain (Fig. 3d), suggesting that the N-domain carries the determinant.

The high similarity of the 3D-structure between the N-domain and C-domain (Fig. 3b) enabled us to create a series of single domain mutants as chimeras of these domains. We identified the determinant for Lewis X modification using these chimeric mutants. The LAMP-1 N-domain, but not chimera B, exhibited Lewis X modification, indicating that the C-terminal half of the N-domain is required for Lewis X modification (Fig. 3d). The series of chimeric mutants enabled us to narrow down the candidate sequence to a 29-amino acid segment from Ile136 to Asn164 (Fig. 3e, f), possessing no N-glycosylation site. A LAMP-1 mutant in which the 29-amino acid sequence of the N-domain was replaced with the corresponding segment of the C-domain resulted in decreased Lewis X modification (Fig. 3g). These data indicate that the contiguous 29-amino acid sequence of LAMP-1 (L29) is responsible for promotion of the FUT9-catalyzed Lewis X modification.

**Embedding the Lewis X modification into model glycoproteins with the L29 sequence.** To examine whether the L29 sequence is sufficient for an encounter with FUT9, we attached this sequence to the C-terminus of erythropoietin (EPO) and fetuin as model glycoproteins and evaluated Lewis X modification (Fig. 4a and Supplementary Fig. 4). The results indicated that the L29 segment evoked a Lewis X modification (Fig. 4b, c). Moreover, using a

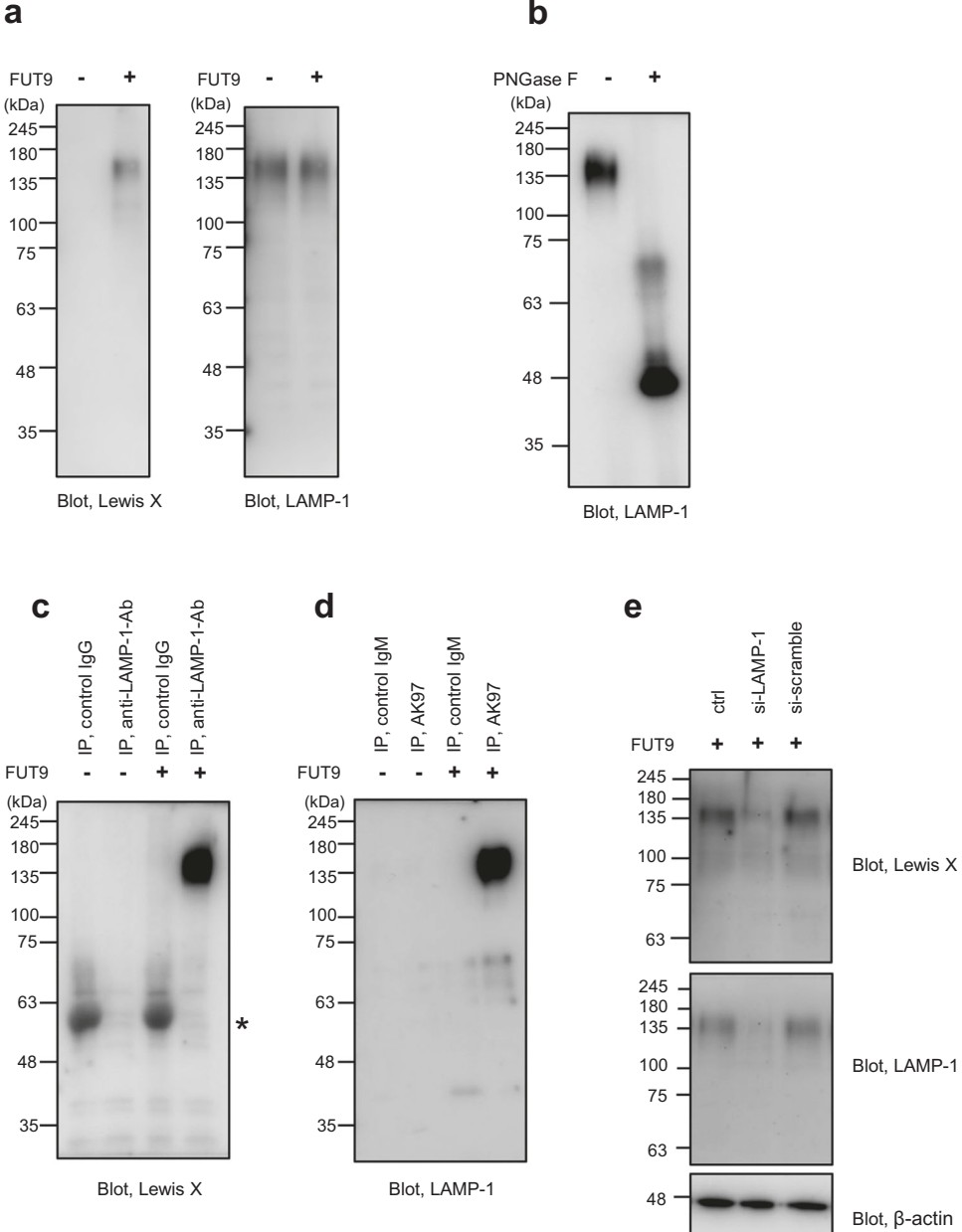

**Fig. 1 FUT9-dependent Lewis X modification of LAMP-1 in CHO-K1 cells. a** Immunoblot analysis of Lewis X modification (left) and LAMP-1 (right) in lysates from CHO-K1 cells transfected with or without FUT9 expression vector using the anti-Lewis X antibody, AK97, and anti-LAMP-1 antibody, respectively. **b** Immunoblot analysis of LAMP-1 in lysate of the CHO-K1 cells expressing FUT9 before and after PNGase F treatment. **c** Immunoblot analysis of Lewis X in cell lysates immunoprecipitated with anti-LAMP-1 antibody and control IgG (IP). The band indicated by asterisk corresponding to control IgG non-specifically detected by the secondary anti-mouse IgM antibody. **d** Immunoblot analysis of LAMP-1 in cell lysates immunoprecipitated with anti-Lewis X and control IgM (IP). **e** Effects of knockdown of LAMP-1 on Lewis X modification in the CHO-K1 cells expressing FUT9. β-actin was used as a loading control.

proximity-dependent biotinylation technique, we observed that recombinant EPO encountered with FUT9 in CHO-K1 cells, depending on the presence of the C-terminal L29 segment (Fig. 5a). Furthermore, we determined whether the L29 segment was effective under cell-free conditions. Recombinant FUT9 and EPO with or without the C-terminal L29 tag were individually produced by CHO-K1 cells, harvested, and then mixed in vitro. Immunoblot analysis of the reaction mixture revealed that recombinant FUT9 preferentially catalyzed the Lewis X modification on L29-tagged EPO, indicating that other cellular components were dispensable for inducing the observed modification (Fig. 5b). Our findings demonstrated that the L29 sequence

derived from the LAMP-1 N-domain is sufficient for interactions of model glycoproteins with FUT9 and their consequent Lewis X modification.

## Discussion

Although LAMP-1 is known as a lysosomal marker, this protein is expressed also on cell surfaces and mediates cell–cell communication through interactions with lectins such as galectin-3 and E-selectin[12,13]. Our previous study has shown that LAMP-1 is a major Lewis X-carrying protein in neural stem cells, and moreover, its Lewis X moieties play an essential role for maintenance

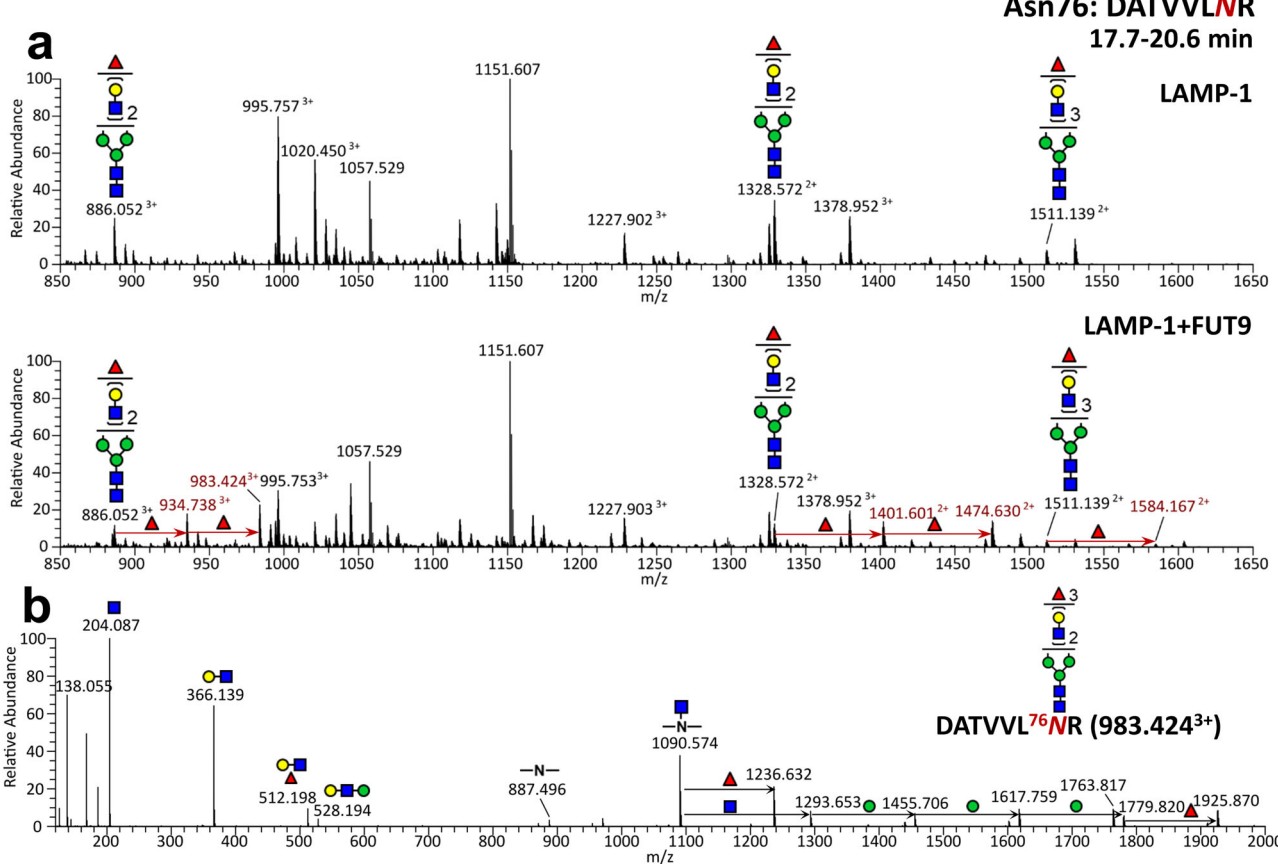

**Fig. 2 Representative MS profiling of site-specific glycosylation of LAMP-1. a** The representative LC-MS spectrum of the glycopeptide containing the Asn76 N-glycosylation site from LAMP-1 produced in wild-type or FUT9-expressing CHO-K1 cells. The peptides containing the Asn76 N-glycosylation site eluted in a time range of 17.7–20.6 min. **b** The representative LC-MS/MS spectrum of the Asn76-containing glycopeptide of LAMP-1 co-expressed with FUT9. The LAMP-1 glycoproteins subjected to the LC-MS measurement were prepared from the gel pieces according to the procedure described in the Methods section. The glycan composition and probable structures were inferred from the few critical fragment ions afforded and the expected range of documented N-glycan structures in the literature. Man, Gal, Fuc, and GlcNAc are represented by symbols according to the Symbol Nomenclature for Glycans (SNFG) (http://www.ncbi.nlm.nih.gov/books/NBK310273/).

of their stemness via activation of Notch signaling[10], while the intracellular function of Lewis X-carrying LAMP-1 remains to be elucidated. We have also demonstrated that the Lewis X modification catalyzed by FUT9 is a prerequisite for this LAMP-1 function in neural stem cells. However, the mechanism of how Lewis X modification occurs exclusively on LAMP-1 remains to be addressed. In this study, we successfully identified a unique amino acid sequence in the N-domain of LAMP-1 that mediates its specific interaction with FUT9 and thereby promotes Lewis X modifications of its N-glycans. Interestingly, this sequence can be embedded into other glycoproteins to induce their Lewis X modifications as we demonstrated for EPO and fetuin. Based on these results, we conclude that the amino acid sequence from LAMP-1 functions as a "Lewis X code", which is deciphered by FUT9 for selective Lewis X modification.

This is contrary to the general view that N-glycosylated glycoproteins interact with glycosyltransferases in the Golgi in promiscuous manners, rendering the N-glycan diversification difficult to predict[2,14,15]. However, in contrast, protein-specific O-glycosylation has been observed in several molecular systems, which is best exemplified by the formation of matriglycans exclusively on α-dystroglycan[16,17]. In this system, several glycosyltransferases coordinate to generate a unique glycan core structure, which is linked to specific threonine residues of the protein and acts as a primer for the elongation of outer α1-

3Xylβ1-3GlcA repeat sequences[18–20]. For the protein-specific unique glycan formation even in a site-specific manner, amino acid sequences around the glycosylation sites are supposed to be recognized by the enzymes involved in these processes. Moreover, amino acid sequence recognition has been reported for POGLUT1, which catalyzes the specific O-glucosylation of the Notch EGF domain[21]. Thus, the protein-selective expression of unique O-glycans can be attributed to amino acid sequence recognition by the specific enzymes involved in the early stage of glycan core formation.

In contrast, there have been few reports on the unique N-glycosylation of a specific protein with a few exceptions, such as polysialylation of neural cell adhesion molecule (NCAM) and HNK-1 modification of tenascin-C[22,23]. Mechanistic insights have been provided for substrate-specific polysialylation: an acidic patch on the surface of the first fibronectin domain of NCAM electrostatically interacts with basic regions of polysialyltransferases[24,25]. The Lewis X code identified in this study contains both basic and acidic residues along with hydrophobic residues and can be embedded into other glycoproteins, suggesting a capability of interacting with FUT9 in a conformation-independent manner. The L29 sequences are conserved among mammals, but not those of non-mammalian species (Supplementary Fig. 5). Therefore, "Lewis X code" we identified might be valid only among mammals. In mouse brain,

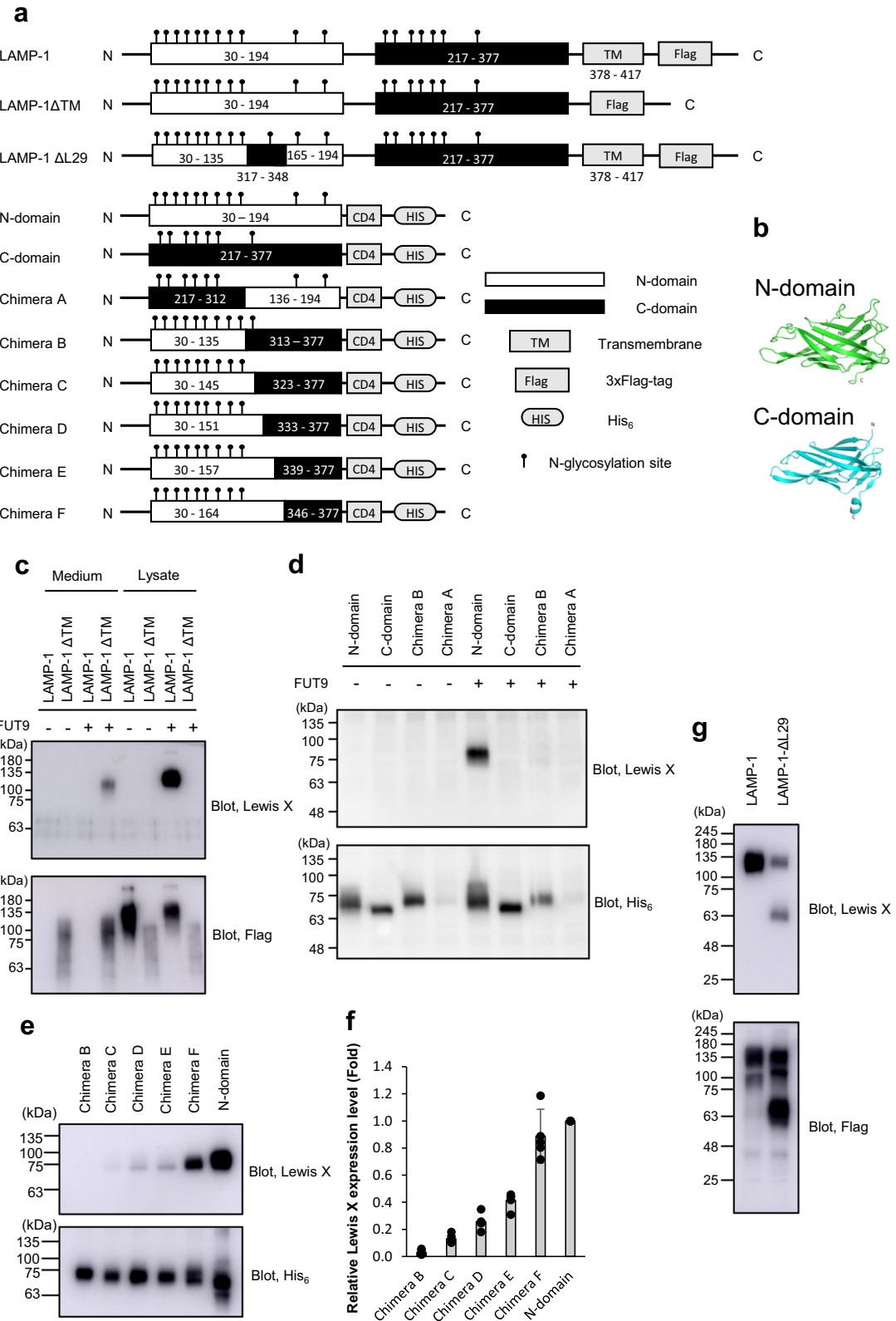

phosphacan, tenascin-C, and L1-CAM have been reported to be modified with Lewis X[10,26,27]. However, we found no sequence similar to the L29 sequence in these glycoproteins. Furthermore, we attempted to find sequence similar to the L29 sequence by conventional BLAST searches but without success. We suppose that a structural view of the interaction mode between FUT9 and the L29 sequence is necessary to identify a critical motif hidden in the sequence. It is also possible that there exist alternative motifs as determinants of Lewis X modifications in those glycoproteins. Moreover, beside the Lewis X code, there may be other hidden codes for specific N-glycosylation in substrate proteins that have not yet been identified.

In glycobiology, glycosylation in cells has been considered to be biased primarily by up- and down-regulation of glycosyltransferases,

**Fig. 3 Identification of the LAMP-1 segment responsible for the FUT9-dependent Lewis X modification. a** Schematic representation of the recombinant proteins used in this study. The recombinant proteins were subjected to immunoblotting after purification using the affinity tag. **b** Three-dimensional structure models of the N- and C-domains of LAMP-1 predicted by AlphaFold 2[35] using an API hosted at the Södinglab based on the MMseqs2 server[36] for multiple sequence alignment. **c** Immunoblot analysis of Lewis X expression on 3xFlag-tagged LAMP-1 and its mutants with or without FUT9 overexpression. **d** Immunoblot analysis of Lewis X expression on the His₆-tagged N/C-domain of LAMP-1 chimeric mutants A and B with or without FUT9 overexpression. **e** Immunoblot analysis of Lewis X expression on the His₆-tagged LAMP-1 N-domain or N/C-chimeric domains with FUT9 overexpression. **f** The density plot (Lewis X/Flag) of relative Lewis X expression levels normalized by Lewis X expression level on the N-domain. Error bars represent the SEM ($n = 3$ independent experiments). **g** Immunoblot analysis of Lewis X expression on 3xFlag-tagged wild-type LAMP-1 and LAMP-1 ΔL29 with FUT9 overexpression.

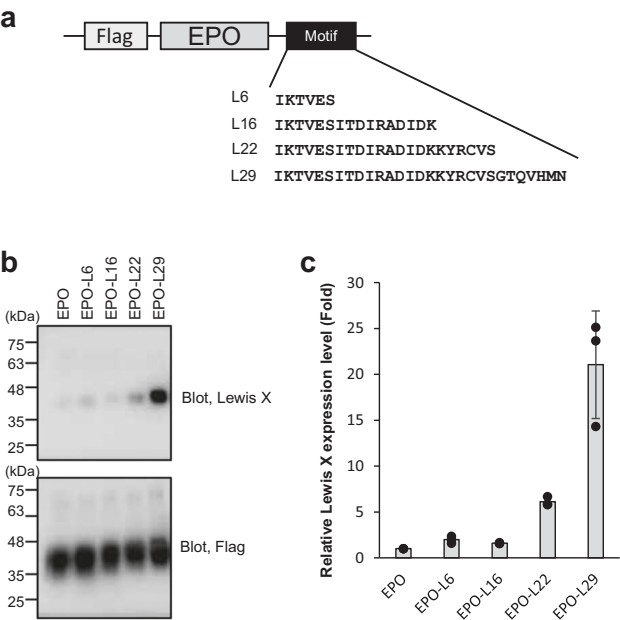

**Fig. 4 FUT9-dependent Lewis X modification of EPO tagged with a LAMP-1-derived peptide. a** Schematic representation of a model glycoprotein, EPO, containing various lengths of amino acid sequence segment derived from LAMP-1. **b** Lewis X expression on 3xFlag-tagged EPO analyzed by immunoblotting using anti-Lewis X and anti-Flag antibodies. **c** The density plot (Lewis X/Flag) of relative Lewis X expression levels normalized by Lewis X expression level on 3xFlag-tagged EPO. Error bars represent the SEM ($n = 3$ independent experiments).

which results in alterations of protein glycosylation profiles in non-selective manners[2,28–30]. However, glycosylation codes offer an alternative, deliberate approach to address protein-specific glycan functions. Although the present study has demonstrated only limited applicability of the Lewis X code for artificial recombinant glycoproteins in vitro and in cells, the embeddable property of glycosylation codes will open up new possibilities for protein engineering and cell engineering.

## Methods

**Recombinant protein expression vectors**. cDNA construction: The open reading frames of human LAMP-1 and human FUT9 were amplified by PCR using cDNA from HEK293T cells as a template. For the expression of FUT9 and LAMP-1, as well as its chimeric or deletion mutants (Fig. 3a), the open reading frame sequences were cloned into the pCMV14-3×FLAG vector (Sigma). The TurboID-fused FUT9 expression vector was constructed by inserting the TurboID coding region between T64 and T65 of the FUT9 coding region of the expression vector flanked with GS linkers (GGGGS x 3). The DNA fragments coding for erythropoietin (EPO) and fetuin were purchased from Fasmac Co. Ltd. and Integrated DNA Technologies, Inc., respectively, and cloned into the pCMV9-3×FLAG vector (Sigma). These proteins were also expressed as N-terminal fusions to a 29-amino acid segment corresponding to the Ile136 to Asn164 region of LAMP-1. The coding region of LAMP-1ΔTM was amplified through inverse PCR utilizing the LAMP-1 expression vector as the template. Expression vectors for the N- and C-domains of LAMP-1

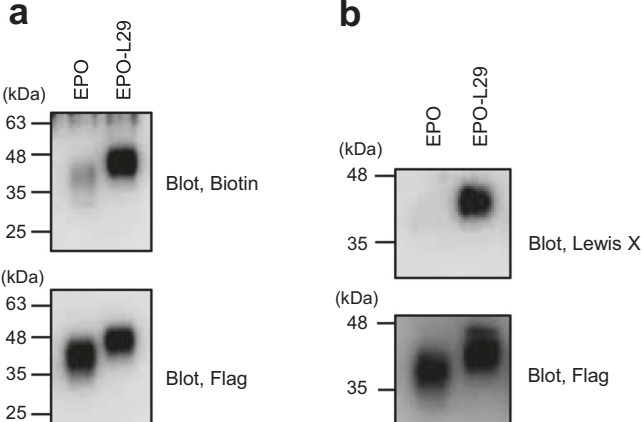

**Fig. 5 Interaction of FUT9 with EPO dependent upon the L29 sequence. a** Proximity-dependent biotinylation of EPO tagged with the L29 sequence by co-expressed FUT9-TurboID in CHO-K1 cells. After proximity labeling, the 3xFlag-tagged EPO glycoproteins with or without L29 sequence (EPO or EPO-L29, respectively) were purified using an anti-Flag M2 Affinity Gel. The purified proteins were detected using avidin-HRP and anti-Flag M2 antibody. **b** FUT9-catalyzed Lewis X formation of desialylated EPO under cell-free conditions. After confirming their full desialylation (Supplementary Fig. 6), the 3xFlag-tagged EPO with or without the L29 sequence (EPO or EPO-L29, respectively) were incubated with FUT9. The reaction mixtures were subjected to immunoblot analysis using anti-Lewis X and anti-Flag antibodies.

(N-domain 1-194, C-domain 1-29 directly joined to 217-377) were produced by the amplification of coding sequence, followed by the insertion at the *NotI/AscI* sites of mammalian LAMP2-bio-His (Addgene; 51861)[31]. The DNA fragments coding for chimeric mutants of LAMP-1 were purchased from Fasmac Co. Ltd and cloned into the pCMV9-3×FLAG vector as shown in Fig. 3.

**Cell culture, transfection, and recombinant protein purification**. CHO-K1 cells were cultured in Dulbecco's modified Eagle's medium (DMEM) supplemented with 10% fetal bovine serum in 5% $CO_2$ at 37 °C. For transfection, cells were grown overnight and transfected using Polyethylenimine "Max" (Polysciences, Inc.) as described previously[32]. At 72 h post-transfection, the recombinant proteins were purified with anti-Flag M2 Affinity Gel (Sigma-Aldrich) or cOmplete™ His-Tag Purification Resin (Roche) from culture medium. Proteins were eluted from each resin with 3X FLAG tag peptide (Sigma-Aldrich) or 500 mM imidazole in PBS. Recombinant membrane proteins were purified with those affinity tags from cell lysate prepared by homogenizing the cultured cells in lysis buffer (20 mM Tris-HCl, pH 7.6, 150 mM NaCl, 1 mM EDTA, 1% Triton X) followed by centrifugation.

**Immunoblotting**. Immunoblot analysis was performed as previously described[10]. Recombinant proteins or cell lysates were subjected to sodium dodecyl sulfate-polyacrylamide gel electrophoresis (SDS-PAGE) and subsequently transferred to polyvinylidene difluoride (PVDF) membranes (Millipore). After blocking with Blocking One solution (Nacalai Tesque), the membranes were incubated with monoclonal mouse anti-Flag M2 antibody (Sigma-Aldrich, F1804) and AK97[33], followed by incubation with respective horseradish peroxidase (HRP)-conjugated secondary antibodies, anti-mouse IgG antibody-HRP (GE Healthcare) and anti-mouse IgM antibody-HRP (ENZO), respectively. The hexahistidine (His₆)-tagged and biotin-labeled recombinant proteins were detected using anti-His₆-Peroxidase (Roche) and HRP-conjugated Streptavidin (Sigma), respectively. The protein bands

were developed with Immobilon Western Chemiluminescent HRP substrate solution (Millipore) and were imaged with an Amersham™ Imager 600 (GE Healthcare). After removing the antibodies or avidin by Western Blot Stripping Buffer (Takara Bio Inc.), the membrane was reproved using different antibodies. For deglycosylation, lysates were incubated with 50 units/mL of PNGase F (New England BioLabs) at 37 °C for 48 h before being subjected to SDS-PAGE. Densitometry analysis was performed using Amersham™ Imager 600 Analysis Software (GE Healthcare).

**Immunoprecipitation**. For immunoprecipitation, the cell lysates were gently agitated with AK97 antibody and control mouse IgM (MBL #M079-3) or anti-LAMP-1 rabbit polyclonal IgG antibody (Abcam #ab24170) and control rabbit IgG (Seikagaku Kogyo Co. #270358) for 15 min, followed by incubation with protein L-Sepharose (ProteNova) or protein G-Sepharose (GE Healthcare), respectively, at 4 °C for 1 h. After washing three times with PBS, immunoprecipitants were subjected to immunoblot analysis.

**RNA interference**. Two small interfering RNAs as 21-mer oligoribonucleotides with a 19-base-pair duplex region and two deoxynucleotide (dtdt) overhangs on the 3′-terminus of each strand, targeting Chinese hamster LAMP-1 (GUU-CUAGCCUGUUUUCCUdtdt and GGUCCAGCGCUUUCAAGGUUdtdt) and the scrambled control (GCUUCUGUUCGUCUCUAUUdtdt) were purchased from NIPPON GENE. CHO-K1 cells were transfected with the FUT9 expression vector, incubated for 24 h, and then transfected with siRNA using Lipofectamine RNAiMAX (Invitrogen) according to the manufacturer's instruction. After 48 h incubation, the cell lysates were collected and subjected to immunoblotting.

**Site-specific glycosylation profiling of LAMP-1**. The gel band containing LAMP-1 was excised and subjected to in-gel digestion sequentially by reduction with 10 mM dithiothreitol at 37 °C for 1 h, alkylation with 50 mM iodoacetamide in 25 mM ammonium bicarbonate buffer for 1 h in the dark at room temperature, destaining with 50% acetonitrile in 25 mM ammonium bicarbonate buffer, and overnight digestion with sequencing-grade trypsin (Promega) and Asp-N (Promega) at 37 °C. The digested products were sequentially extracted with distilled water, 1% formic acid, and 50% acetonitrile/1% formic acid, dried, redissolved in 0.1% formic acid, and further cleaned up on a ZipTip C18 (Merck Millipore) column before analysis. The peptide mixtures were analyzed by nanospray LC-MS/MS on an Orbitrap Fusion Lumos Tribrid (Thermo Fisher Scientific) coupled to an Easy-nLC 1200 (Thermo Fisher Scientific). Peptide mixtures were loaded onto an Acclaim PepMap RSLC 25 cm × 75 μm i.d. column (Thermo Fisher Scientific) and separated at a flow rate of 300 nL/min using a gradient of 5 to 40% solvent B (80% acetonitrile with 0.1% formic acid) for 60 min. Solvent A was 0.1% formic acid in water. The parameters used for MS and MS/MS data acquisition under the HCD product ion trigger CID mode were: top speed mode with 3-s cycle time; FTMS: scan range ($m/z$) = 450–1700; resolution = 120 K; AGC target = $2 \times 10^5$; maximum injection time = 50 ms; monoisotopic precursor selection on; including charge state 2–10; dynamic exclusion after one time and exclusion for 35 s with 15-ppm tolerance. FTMSn (HCD): isolation mode = quadrupole; isolation window = 2; collision energy 28%; resolution = 30 K; AGC target = $5 \times 10^4$; maximum injection time = 65 ms; HCD production ions $m/z$ 138.0545, 204.0867, or 366.1396 ($z = 1$) and the top 20 product ions were used to trigger CID; ITMSn (CID): isolation mode = quadrupole; isolation window = 2; collision energy = 30%; AGC target = $1 \times 10^4$; ion trap scan rate = rapid. The probable glycopeptide hits were firstly identified by Byonic software (Protein Metrics Inc., version 4.0.12) using the following parameters: search against default N-glycan 132 database removing N-glycans smaller than trimannosyl core structure (Hex-NAc$_2$Hex$_3$), using semi-specific cleavages at K, R (C-terminal) and D (N-terminal) residues, allowing up to 2 missed cleavages, with the precursor ion mass tolerance set at 5 ppm and the fragment ion mass tolerance at 10 ppm. Variable common modifications considered were carbamidomethylation (+57.0215 Da, at C), oxidation (+15.9949 Da, at M). Variable rare modification considered was deamidation (+0.9840 Da, at N and Q). The criteria used in additional manual filtering of positive matches were PEP 2D < 0.01, score >150. HCD and CID MS$^2$ datasets were also manually checked by filtering the candidate glycopeptide spectra based on the presence of MS$^2$ ions corresponding to the expected tryptic peptide core containing the target N-glycosylation sites and/or peptide fragment ions, and then manually interpreted and assigned as described in the "Results".

**TurboID-based proximity labeling**. Proximity labeling using TurboID was performed as previously described[34]. Briefly, biotin was added to the culture medium at 48 h after co-transfection of FUT9-TurboID and Flag-tagged EPO with or without the L29 sequence. After incubation for 18 h, the Flag-tagged recombinant proteins were purified from the medium on anti-Flag M2 Affinity Gels, subjected to SDS-PAGE, and electroblotted to protein-blotting membrane. The proteins were detected using avidin-HRP (Thermo Scientific) and anti-Flag M2.

**In vitro enzymatic activity assay of FUT9**. For desialylation, ~1 μg/ml of EPO with or without the L29 sequence were dissolved in 0.1 M MES buffer, pH 6.5, containing 150 mM NaCl, and then incubated with 250 μunits/mL of sialidase from *Arthrobacter ureafaciens* (Nacalai tesque) at 37 °C for 1 h. Desialylation was confirmed based on mobility shift in SDS-PAGE (Supplementary Fig. 6). Then, the

desialylated glycoprotein solution was mixed with an equivolume of the same buffer solution containing 2 μg/mL of 6-deoxy-β-L-galactopyranosylguanosine 5′-diphosphate-fucose (Sigma), and ~2 μg/ml of FUT9, 40 mM MnCl$_2$, and 0.02% Triton X-100, incubated at 37 °C for 1 h, and then subjected to immunoblot analysis to detect Lewis X modification.

**Statistics and reproducibility**. Statistical analysis was performed by two-tailed unpaired *t*-test with Welch's correction to evaluate significance using Microsoft Excel (Microsoft Corporation). Statistical significance was set at $P < 0.05$. The sample size is shown in the figure legends.

**Reporting summary**. Further information on research design is available in the Nature Research Reporting Summary linked to this article.

## Data availability

The mass spectrometric data have been deposited to the Mass Spectrometry Interactive Virtual Environment (MassIVE) with the dataset ID: MSV000089274. Unedited blot images are presented in Supplementary Figs. 7–11. The statistical source data for all the graphs presented in figures are available as Supplementary Data 3.

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

## Acknowledgements
We thank Kiyomi Senda (Nagoya City University) for help in antibody preparation from hybridoma cells. This work was supported in part by JSPS Grant-in-Aid for JSPS Research Fellow (to T.S.), Grants-in-Aid for Scientific Research (Grant Numbers JP20J23703 to T.S. and JP17H06414 and JP21H02625 to H.Y. and JP20K21495, JP24249002 and JP25102008 to K.K.), JST-CREST, Grant Number JP MJCR21E3 to K.K., Joint Research by Exploratory Research Center on Life and Living Systems (ExCELLS) (ExCELLS program Nos. 21-302, 22EXC304, and 22EXC318) and an Academia Sinica Investigator Award grant AS-IA-105-L02 to K.H.K.). Mass spectrometry data were acquired at the Academia Sinica Common Mass Spectrometry Facilities for Proteomics and Protein Modification Analysis (supported by grant AS-CFII-108-107).

## Author contributions
Conceived and designed experiments: H.Y. and K.Kato. Performed the biochemical experiments: T.S. and H.Y. Performed the MS experiments: T.S., C.K., and K.Khoo. Wrote the paper: T.S., H.Y., C.K., K.Khoo, and K.Kato.

## Competing interests
The authors declare no competing interests.
