## [Peer Review File · Communications Biology]

Reviewers' comments:

Reviewer #1 (Remarks to the Author):

The authors describe the finding of an embeddable molecular code for Lewis X motif/SSEA-1 on lysosome-associated membrane protein 1(LAMP-1) based on results on mutation experiments, blotting assays and mass spectrometry. The finding is interesting and could be useful as a generic tool to introduce this epitope on any glycoproteins. However, here are my questions.

1. What is the biological function of this glycan epitope on LAMP1? LeX is known to be a biomarker on cell surface. The appearance of LeX inside of a cell on LAMP1 does not make sense unless that there is a receptor/reader protein inside of cells that interacts with this glycan epitope.

2. The glycan outer branches are usually terminated with sialylation and fucosylation. These two modifications frequently have intricate relationship. It is known that sialylation can prevent fucosylation in LeX generation. In other words, if the molecular code does exist, it should not only promote fucosylation by FUT9 on LAMP1 but also likely prevent sialylation on LAMP1. For examples, in Fig. 5 and Supplemental Fig.3, no LeX was found on EPO and Fetuin, which could be due to that sialylation on these proteins prevent fucosylation by FUT9. It is better for the authors to address this issue.

3. If the molecular code does exist, it must also have a mechanism to attract FUT9. To demonstrate this idea, it is better to do the cell free experiment on purified asialoproteins such as asialofetuin so that to avoid the sialylation issue mentioned in point 2.

4. In the last paragraph of Page 4. Chimera 1 and chimera 2 were not indicated in Fig.3

5. LAMP1 was mentioned as an 80 kDa protein in their original paper (Glycobiology vol. 20 no. 8 pp. 976–981, 2010) but was mentioned as a 135 kDa protein in this manuscript. It is better to address this large discrepancy.

Reviewer #2 (Remarks to the Author):

The manuscript submitted from Dr. Kato, et al. describes the identification and verification of a pseudo-"consensus motif" of amino acids within a protein sequence that is a deterministic factor for Lewis-X decoration of N-linked glycans on specific proteins. The work built on a previous study from the same lab which identified a 29-aa N-terminal fragment of the LAMP-1 protein which interacts specifically with the FUT9 glycosyltransferase to overexpress a Lewis-X motif.

The work is original and convincing. The experiments are controlled, and clear. The approach uses various methodology (antibody-specific, westerns; gel-extracted LC-MS of glycopeptides) to support the claims.

Overall, the results are important to the glycosciences community, as they suggest a much more nuanced, targeted approach of evolution to our biology in regards to glycosylation of proteins. The work may also have some impact on the biomanufacturing community, where controlling protein glycosylation during culture is advantageous.

There are some limitations to the work, which might be more clearly stated in the text. First, all of this work was done in cell culture or in cell-free expression systems, using mutated, non-native structures with various modifications and affinity tags. Although there is no proven reason that this mechanism wouldn't apply in a natural system (in vivo), this work does not prove that it does. The authors have not overstated this work, but the limitation could be noted.

For glycoprotein expression, this does not matter, but for studying disease states associated with Lewis-X motifs, or for extrapolating to other consensus motif-driven glycosylation, it should be noted that cell culture is not always a perfect representation.

Second, the authors state that "These data indicate that the contiguous 29-amino acid sequence of LAMP-1 (L29) is indispensable for its FUT9-dependent Lewis X modification" in the second section of the Results. This might be mostly true, but it is perhaps overstated. It is not clearly proven that the entire 29-amino acid fragment is necessary, but it has been proven to be sufficient. More

studies could be done that would clarify this, but those experiments are not essential for this review. Additionally, the 29-aa from LAMP-1 C-domain was also shown sufficient (but not necessary), although to a lesser impact. It might also be interesting to show whether or not the the N-terminal amino acid sequence, or smaller fragments of it, are evolutionarily conserved, and in what proteins. If this was done it should be stated in the paper, if it was not done it is quite easy to do – but again, this is not essential to this reviewer.

Thirdly, although the data is very convincing and the mechanism has been shown for three independent proteins (fetuin, EPO, LAMP-1), the authors should not extrapolate the findings too much. The last sentence of the Results “Our findings demonstrated that the L29 sequence derived from the LAMP-1 N-domain is necessary and sufficient for the interaction of its carrier proteins with FUT9 and its consequent Lewis X modification” could be clarified to be specifically for the 3 glycoproteins for which data is provided.

Overall, a very nice paper, great data to support it, and very interesting study. I wish the authors the best of luck moving this work forward to more motifs, and more applications.

Reviewer #3 (Remarks to the Author):

In this paper, Kato and coworkers describe how they have discovered a peptide within the protein LAMP-1 that mediates fucosylation of some of the protein's N-glycans by the fucosyltransferase FUT-9, generating Lewis-X epitopes. The results are novel and I do not doubt the validity of the presented data. However, I do think that the authors have been a bit hasty drawing conclusions and some of the claims that are being made are not fully supported by the data. In addition, I think the significance of the paper, and its impact on advancing understanding in the field, could be massively improved by some deeper analysis of the findings and generated constructs (without this needing to be a huge amount of work).

My main problem with the conclusions of the paper involves the claim that the amino acid sequence is indispensable for fucosylation; Figure 3G shows that deletion of the peptide in LAMP-1 gives a lower, yet still visible, amount of Lewis-X signal. While the experiments with EPO and fetuin indeed prove that the peptide is sufficient for fucosylation by FUT-9, there is no evidence that the peptide is absolutely necessary.

Of less importance to the overall conclusions of the paper, but still important, I do not agree with the author's assessment of LAMP-1 being the only protein modified with Lewis-X. The observed band in lysates (Fig 1A) is a broad smear and it is impossible to say whether that is a single protein or not. The IP-blotting experiments (Fig 1B/C) provide evidence that LAMP-1 carries Lewis-X, but they do not exclude the possibility of other proteins carrying the same epitope. In HEK and COS cells (Fig S1) there are definitely multiple bands visible in the Lewis-X blot; and in COS cells, the Lewis-x signal does not overlap well with the LAMP blot. While this does affect the identified recognition motif in LAMP-1, the conclusions need to be toned down accordingly. Alternatively, a LAMP-1 knockdown experiment could provide strong evidence for LAMP-1 being the exclusive FUT-9 target in these cells (or not).

In terms of strengthening the paper, it would be very interesting to find out whether the same (or a similar) peptide motif can be found in other proteins, including known FUT-9 targets (e.g. those described in literature); this could be relatively easily done by a database search and would add valuable extra information.

In addition, when the authors have identified the peptide that mediates Lewis-X formation, they unfortunately do not go back to confirm the fucosylation status of the LAMP-1 constructs by LC-MS/MS analysis. Doing so, for example using the delta 29 construct, would provide additional evidence for the paper's main findings and, perhaps more interestingly, it would also provide

information on which fucosylation sites - out of those identified on page 4 - are affected by the deletion of the peptide. This is an interesting question especially since the identified peptide itself is not shown to be modified by FUT-9 (something worth mentioning in the text!).

Finally, there is a statement for which I can't find the corresponding data: 'Chimera 1, but not chimera 2, exhibited Lewis X modification, indicating that the C-terminal half of the N-domain is required for Lewis X modification.' The naming of the chimeras does not correspond to that in the figure, but I assume that chimera 1 means chimera A. However, this construct does not show any signal on the blot: it looks like this is a problem of the protein not being expressed well or being unstable, as there is no HIS signal either. Please clarify and add missing data.

Minor corrections/questions:

- Why does the negative control in Figure 1B (IgG IP without antibody) show a prominent band on the Lewis-X blot, whereas the LAMP-1 antibody IP sample does not?
- In Fig 1A, the observed signal runs higher than 135 kDa and is a very broad smear rather than a defined band. Can the authors provide a reference for the expected apparent molecular weight of LAMP-1 on sds-page/blot in this cell line?
- 'We found that recombinant LAMP-1 underwent a Lewis X modification in FUT9-overexpressing CHO-K1 cells'. It is unclear what data this is based on.
- Site-specific N-glycosylation profiling: was the protein purified or was it excised from a gel? This is unclear (main text suggests purification but methods section does not).
- In the description of figure 3 (and the figure caption), it is unclear whether the overexpressed proteins were first purified and then analysed or whether these blots come from whole cell lysates/media.
- Likewise, in the methods section, there is insufficient information on the overexpression and purification of the various constructs: e.g. were these proteins excreted or were they purified from lysates? What was the lysis buffer? Which of the proteins were purified by anti-FLAG and which by HIS tag resin (and where is the HIS tag? This is not mentioned in the vectors..)?
- Also missing is a description of the densitometry analysis.
- The following sentence in the intro is a bit confusing and could do with some rephrasing: 'Cellular N-glycosylation ... produced in the cell': I don't understand why promiscuity leads to prevalent effects among glycoproteins.

Response to Reviewers' comments:

Reviewer #1 (Remarks to the Author):

The authors describe the finding of an embeddable molecular code for Lewis X motif/SSEA-1 on lysosome-associated membrane protein 1(LAMP-1) based on results on mutation experiments, blotting assays and mass spectrometry. The finding is interesting and could be useful as a generic tool to introduce this epitope on any glycoproteins. However, here are my questions.

1. What is the biological function of this glycan epitope on LAMP1? LeX is known to be a biomarker on cell surface. The appearance of LeX inside of a cell on LAMP1 does not make sense unless that there is a receptor/reader protein inside of cells that interacts with this glycan epitope.

Although LAMP-1 is known as a lysosomal marker, this protein is expressed also on cell surfaces and mediates cell-cell communication through interactions with lectins such as galectin-3 and E-selectin [Int. J. Cancer. (1998) **75**, 105-111 and Int. J. Oncol. (2000) **16**, 347-353]. Our previous study has shown that LAMP-1 is a major Lewis X-carrying protein in neural stem cells and play an essential role for maintenance of their stemness via activation of Notch signaling [Glycobiology. (2010) **20**, 976-81]. We have also demonstrated that the Lewis X modification catalyzed by FUT9 is a prerequisite for this LAMP-1 function in neural stem cells. We described these points in the text (p. 5, lines 8-33).

2. The glycan outer branches are usually terminated with sialylation and fucosylation. These two modifications frequently have intricate relationship. It is known that sialylation can prevent fucosylation in LeX generation. In other words, if the molecular code does exist, it should not only promote fucosylation by FUT9 on LAMP1 but also likely prevent sialylation on LAMP1. For examples, in Fig. 5 and Supplemental Fig.3, no LeX was found on EPO and Fetuin, which could be due to that sialylation on these proteins prevent fucosylation by FUT9. It is better for the authors to address this issue.

We thank the reviewer for the constructive comment. In the experiments, we used desialylated glycoproteins prepared by a sialidase treatment because Lewis X formation is prevented by sialylation as the reviewer pointed out. We apologize for not clearly stating this point in the original manuscript. In the revised manuscript, we modified the descriptions in the Figure 5 legends and the Methods section (p. 10, lines 6-12) to explain this point.

3. If the molecular code does exist, it must also have a mechanism to attract FUT9. To demonstrate this idea, it is better to do the cell free experiment on purified asialoproteins such as asialofetuin so that to avoid the sialylation issue mentioned in point 2.

As mentioned above, we have actually tested *in vitro* fucosylation by FUT9 using the desialylated forms of acceptor glycoproteins. We consider the direct interaction between FUT9 and the molecular code as the mechanism.

4. In the last paragraph of Page 4. Chimera 1 and chimera 2 were not indicated in Fig.3

We apologize for the careless mistake, which was corrected in the revised manuscript (p. 5, line 3).

5. LAMP1 was mentioned as an 80 kDa protein in their original paper (Glycobiology vol. 20 no. 8 pp. 976–981, 2010) but was mentioned as a 135 kDa protein in this manuscript. It is better to address this large discrepancy.

We thank the reviewer for the insightful comment. The discrepancy is due to glycosylation differences. To confirm this point, the cell lysate treated with PNGase F was subjected to immunoblotting with the anti-LAMP-1 antibody. The result indicated that the deglycosylated LAMP-1 migrated as a 40-kDa protein, which is consistent with the observation in our previous study. We showed the data in the revised manuscript (p. 3, lines 33-35) by adding Figure 1B.

Reviewer #2 (Remarks to the Author):

The manuscript submitted from Dr. Kato, et al. describes the identification and verification of a pseudo-"consensus motif" of amino acids within a protein sequence that is a deterministic factor for Lewis-X decoration of N-linked glycans on specific proteins. The work built on a previous study from the same lab which identified a 29-aa N-terminal fragment of the LAMP-1 protein which interacts specifically with the FUT9 glycosyltransferase to overexpress a Lewis-X motif. The work is original and convincing. The experiments are controlled, and clear. The approach uses various methodology (antibody-specific, westerns; gel-extracted LC-MS of glycopeptides) to support the claims. Overall, the results are important to the glycosciences community, as they suggest a much more nuanced, targeted approach of evolution to our biology in regards to glycosylation of proteins. The work may also have some impact on the biomanufacturing community, where controlling protein glycosylation during culture is advantageous.

There are some limitations to the work, which might be more clearly stated in the text. First, all of this work was done in cell culture or in cell-free expression systems, using mutated, non-native structures with various modifications and affinity tags. Although there is no proven reason that this mechanism wouldn't apply in a natural system (in vivo), this work does not prove that it does. The authors have not overstated this work, but the limitation could be noted. For glycoprotein expression, this does not matter, but for studying disease states associated with Lewis-X motifs, or for extrapolating to other consensus motif-driven glycosylation, it should be noted that cell culture is not always a perfect representation.

We concur with the reviewer. We reworded “in vivo” to “in cells” (p. 2, line 8) and mentioned that the present study has demonstrated only limited applicability of the Lewis X code for artificial recombinant glycoproteins in vitro and in cells (p. 7, lines 2-3). For clarifying that we are using recombinant proteins in this study, we added sentences at the last part of main text (p. 7, line 3). We also provided information of the modifications of recombinant proteins, including affinity tags, in the schematic diagrams in Figure 4 and the captions of Figures 2, 4 and 5.

Second, the authors state that “These data indicate that the contiguous 29-amino acid sequence of LAMP-1 (L29) is indispensable for its FUT9-dependent Lewis X modification” in the second section of the Results. This might be mostly true, but it is perhaps overstated. It is not clearly proven that the entire 29-amino acid fragment is necessary, but it has been proven to be sufficient. More studies could be done that would clarify this, but those experiments are not essential for this review. Additionally, the 29-aa from LAMP-1 C-domain was also shown sufficient (but not necessary), although to a lesser impact.

We agree with the reviewer on this point as well. Accordingly, we modified the sentence to clarify the point: the L29 sequence is sufficient (or responsible) for promotion of the FUT9-catalyzed LAMP-1 formation (p. 5, line 10 and line 24).

It might also be interesting to show whether or not the N-terminal amino acid sequence, or smaller fragments of it, are evolutionarily conserved, and in what proteins. If this was done it should be stated in the paper, if it was not done it is quite easy to do – but again, this is not essential to this reviewer.

We investigated the evolutionary conservation of the L29 sequence. This sequence is highly conserved among mammals but not those of non-mammalian species such as chicken. Therefore, the Lewis X code we identified might be valid among mammals. We provided the information in the text (p. 6, lines 25-26) with Supplementary Figure 5 in the revised manuscript.

Thirdly, although the data is very convincing and the mechanism has been shown for three independent proteins (fetuin, EPO, LAMP-1), the authors should not extrapolate the findings too much. The last sentence of the Results “Our findings demonstrated that the L29 sequence derived from the LAMP-1 N-domain is necessary and sufficient for the interaction of its carrier proteins with FUT9 and its consequent Lewis X modification” could be clarified to be specifically for the 3 glycoproteins for which data is provided.

As per reviewer’s comments, the sentence was modified for clarity in the revised manuscript (p. 5, lines 23-25) : “Our findings demonstrated that the L29 sequence derived from the LAMP-1 N-domain is sufficient for interactions of model glycoproteins with FUT9 and their consequent Lewis X modification.”

Overall, a very nice paper, great data to support it, and very interesting study. I wish the authors the best of luck moving this work forward to more motifs, and more applications.

We are deeply grateful for the positive evaluation and encouraging comments.

Reviewer #3 (Remarks to the Author):

In this paper, Kato and coworkers describe how they have discovered a peptide within the protein LAMP-1 that mediates fucosylation of some of the protein’s N-glycans by the fucosyltransferase FUT-9, generating Lewis-X epitopes. The results are novel and I do not doubt the validity of the presented data. However, I do think that the authors have been a bit hasty drawing conclusions and some of the claims that are being made are not fully supported by the data. In addition, I think the significance of the paper, and its impact on advancing understanding in the field, could be massively improved by some deeper analysis of the findings and generated constructs (without this needing to be a huge amount of work).

My main problem with the conclusions of the paper involves the claim that the amino acid sequence is indispensable for fucosylation; Figure 3G shows that deletion of the peptide in LAMP-1 gives a lower, yet still visible, amount of Lewis-X signal. While the experiments with EPO and fetuin indeed prove that the peptide is sufficient for fucosylation by FUT-9, there is no evidence that the peptide is absolutely necessary.

We appreciate the reviewer for the insightful comment. Indeed, it has been reported that FUT9 is capable of catalyzing fucosylation of *N*-acetylglucosamine in *in vitro* condition where the amount of enzyme to substrates is high enough [FEBS Lett. (1999) **462**, 289-94]. Together with the present data, we could say that the L29 sequence is not essential but facilitates the Lewis X formation. We modified the text in revised manuscript to clarify that the L29 sequence is sufficient (or responsible) for promotion of the FUT9-catalyzed LAMP-1 formation (p. 5, line 10 and line 24).

Of less importance to the overall conclusions of the paper, but still important, I do not agree with the author's assessment of LAMP-1 being the only protein modified with Lewis-X. The observed band in lysates (Fig 1A) is a broad smear and it is impossible to say whether that is a single protein or not. The IP-blotting experiments (Fig 1B/C) provide evidence that LAMP-1 carries Lewis-X, but they do not exclude the possibility of other proteins carrying the same epitope. In HEK and COS cells (Fig S1) there are definitely multiple bands visible in the Lewis-X blot; and in COS cells, the Lewis-x signal does not overlap well with the LAMP blot. While this does affect the identified recognition motif in LAMP-1, the conclusions need to be toned down accordingly. Alternatively, a LAMP-1 knockdown experiment could provide strong evidence for LAMP-1 being the exclusive FUT-9 target in these cells (or not).

We thank the reviewer for useful comment. As per the reviewer's suggestion, we performed the LAMP-1 knockdown experiment. The result indicated that the Lewis X modification was abolished by knockdown of LAMP-1, demonstrating that LAMP-1 is the carrier of the Lewis X glycotope. We added these data in the revised manuscript (p. 4, lines 7-8 and Figure 1E).

In terms of strengthening the paper, it would be very interesting to find out whether the same (or a similar) peptide motif can be found in other proteins, including known FUT-9 targets (e.g. those described in literature); this could be relatively easily done by a database search and would add valuable extra information.

We thank the reviewer for the constructive comment. In mouse brain, phosphacan, tenascin-C, and L1-CAM have been reported to be modified with Lewis X [J Biol Chem. (2013) **288**, 16538-16545, Glycobiology. (2010) **20**, 976-81 and Glycobiology. (2015) **25**, 376-85]. However, we found no sequence similar to the L29 sequence in these glycoproteins. We also performed a conventional BLAST search but could not identify any sequence similar to it. We suppose that a structural view of the interaction mode between FUT9 and the L29 sequence is necessary to identify a critical motif hidden in the sequence. It is

also possible that there exist alternative motifs as determinants of Lewis X modifications in those glycoproteins. We described these points in the revised manuscript (p. 6. lines 25-33).

In addition, when the authors have identified the peptide that mediates Lewis-X formation, they unfortunately do not go back to confirm the fucosylation status of the LAMP-1 constructs by LC-MS/MS analysis. Doing so, for example using the delta 29 construct, would provide additional evidence for the paper's main findings and, perhaps more interestingly, it would also provide information on which fucosylation sites - out of those identified on page 4 - are affected by the deletion of the peptide. This is an interesting question especially since the identified peptide itself is not shown to be modified by FUT-9 (something worth mentioning in the text!).

We thank the reviewer for the constructive comment. Unfortunately, we could not carry out a site-specific N-glycosylation analysis of the L29-deleted mutant because it was unstable and easily degraded as shown in Figure 3G. The comparison of site-specific N-glycosylation profiles of LAMP-1 with and without co-expression of FUT9 indicated that the FUT9-catalyzed Lewis X modification occurred in a non-site-specific fashion. Therefore, we expect that deletion of the FUT9-recognized motif results in decrease in fucosylation levels uniformly at these sites. In addition, the identified peptide itself has no N-glycosylation site and therefore is expected not to be modified by FUT9. We described this point in the revised manuscript (p.5, lines 6-7).

Finally, there is a statement for which I can't find the corresponding data: 'Chimera 1, but not chimera 2, exhibited Lewis X modification, indicating that the C-terminal half of the N-domain is required for Lewis X modification.' The naming of the chimeras does not correspond to that in the figure, but I assume that chimera 1 means chimera A. However, this construct does not show any signal on the blot: it looks like this is a problem of the protein not being expressed well or being unstable, as there is no HIS signal either. Please clarify and add missing data.

We apologize for the careless mistake, which was corrected in the revised manuscript (p. 5, line 3).

Minor corrections/questions:

- *Why does the negative control in Figure 1B (IgG IP without antibody) show a prominent band on the Lewis-X blot, whereas the LAMP-1 antibody IP sample does not?*

The prominent band corresponding to control IgG with non-specific reaction with secondary antibody, which was explained in the Figure 1C legend.

• In Fig 1A, the observed signal runs higher than 135 kDa and is a very broad smear rather than a defined band. Can the authors provide a reference for the expected apparent molecular weight of LAMP-1 on sds-page/blot in this cell line?

LAMP-1 is a 40-kDa protein with 17 N-glycosylation sites. The band smearing with low migration was ascribed to the heterogeneous glycosylation at these sites. We conducted a PNGase F treatment of the cell lysate, which was then subjected to immunoblotting with the anti-LAMP-1 antibody. The result indicated that the deglycosylated LAMP-1 migrated as a 40-kDa protein in SDS-PAGE. We showed the data in the revised manuscript (p. 3, lines 33-35) by adding Figure 1B.

• 'We found that recombinant LAMP-1 underwent a Lewis X modification in FUT9-overexpressing CHO-K1 cells'. It is unclear what data this is based on.

In replying to this reviewer's comment, we added Supplementary Figure 2 explicitly showing that recombinant LAMP-1 underwent a Lewis X formation in the cells (p. 4, line 12).

• Site-specific N-glycosylation profiling: was the protein purified or was it excised from a gel? This is unclear (main text suggests purification but methods section does not).

The N-glycosylation profiles were obtained from the gel band corresponding to LAMP-1. As per the reviewer's comment, we provided the information in the Figure 2 legend.

• In the description of figure 3 (and the figure caption), it is unclear whether the overexpressed proteins were first purified and then analysed or whether these blots come from whole cell lysates/media.

As per the reviewer's comment, we modified the sentence in the Figure 3 legend.

• Likewise, in the methods section, there is insufficient information on the overexpression and purification of the various constructs: e.g. were these proteins excreted or were they purified from lysates? What was

the lysis buffer? Which of the proteins were purified by anti-FLAG and which by HIS tag resin (and where is the HIS tag? This is not mentioned in the vectors..)?

We appreciate the reviewer's careful remarks. As per the comments, we provided the information in the revised manuscript. We modified the sentence in the Methods section and clarify whether recombinant proteins were purified from medium or lysates. (p. 7, lines 30-35) We added description about the lysis buffer also in Methods (p. 7, lines 35-36). The protocols of purification of recombinant proteins were clearly described by adding the sentence "The recombinant proteins were subjected to immunoblotting after purification using the affinity tag." in the Figure 3A legend (p. 16, lines 13-14). The His₆-tag was fused to the C-terminus of each recombinant protein shown in Figure 3A. Also, we mentioned the subcloning process in the Methods section (p. 7, lines 20-23).

- *Also missing is a description of the densitometry analysis.*

We added a description about densitometry analysis in the last line in the immunoblotting section (p. 8, lines 16-17).

- *The following sentence in the intro is a bit confusing and could so with some rephrasing: 'Cellular N-glycosylation ... produced in the cell': I don't understand why promiscuity leads to prevalent effects among glycoproteins.*

We rephrased the sentence by replacing 'prevalent among' with 'non-selective for' (p. 3, line 9).

Reviewers' comments:

Reviewer #1 (Remarks to the Author):

1. What is the biological function of this glycan epitope on LAMP1? LeX is known to be a biomarker on cell surface. The appearance of LeX inside of a cell on LAMP1 does not make sense unless that there is a receptor/reader protein inside of cells that interacts with this glycan epitope. Although LAMP-1 is known as a lysosomal marker, this protein is expressed also on cell surfaces and mediates cell-cell communication through interactions with lectins such as galectin-3 and E-selectin [Int. J. Cancer. (1998) 75, 105-111 and Int. J. Oncol. (2000) 16, 347-353]. Our previous study has shown that LAMP-1 is a major Lewis X-carrying protein in neural stem cells and play an essential role for maintenance of their stemness via activation of Notch signaling [Glycobiology. (2010) 20, 976-81]. We have also demonstrated that the Lewis X modification catalyzed by FUT9 is a prerequisite for this LAMP-1 function in neural stem cells. We described these points in the text (p. 5, lines 8-33).

Comments: Based on the above information, we must conclude that the presence of Lewis X epitopes on LAMP-1 inside of cell has no known biological functions. The functions that the author mentioned belong to cell membrane LAMP-1.

2. The glycan outer branches are usually terminated with sialylation and fucosylation. These two modifications frequently have intricate relationship. It is known that sialylation can prevent fucosylation in LeX generation. In other words, if the molecular code does exist, it should not only promote fucosylation by FUT9 on LAMP1 but also likely prevent sialylation on LAMP1. For examples, in Fig. 5 and Supplemental Fig.3, no LeX was found on EPO and Fetuin, which could be due to that sialylation on these proteins prevent fucosylation by FUT9. It is better for the authors to address this issue.

We thank the reviewer for the constructive comment. In the experiments, we used desialylated glycoproteins prepared by a sialidase treatment because Lewis X formation is prevented by sialylation as the reviewer pointed out. We apologize for not clearly stating this point in the original manuscript. In the revised manuscript, we modified the descriptions in the Figure 5 legends and the Methods section (p. 10, lines 6-12) to explain this point.

Comments: If desialylated glycoproteins were used for FUT9 modifications, the evidence of desialylation is better to be presented as well. Desialylation usually result in gel mobility shift in SDS-PAGE. Without this evidence, it is hard to make the conclusion that L29 sequence is the major factor for regulating FUT9 modification. Besides, Fig 5A is a blot of biotin and anti-Flag, but not anti-Lewis X and anti-Flag as indicated in the figure legend.

3. If the molecular code does exist, it must also have a mechanism to attract FUT9. To demonstrate this idea, it is better to do the cell free experiment on purified asialoproteins such as asialofetuin so that to avoid the sialylation issue mentioned in point 2.

As mentioned above, we have actually tested in vitro fucosylation by FUT9 using the desialylated forms of acceptor glycoproteins. We consider the direct interaction between FUT9 and the molecular code as the mechanism.

Comments: In vitro test on desialylated glycoprotein will be a good experiment. However, there is no evidence of desialylation presented. Desialylation on fetuin will result an obvious gel mobility shift, which can be easily demonstrated, for example. Comparing FUT9 modifications on glycoproteins with and without desialylation is critical to support the claims on L29 sequence, simply because that sialylation can prevent FUT9 modification.

4. In the last paragraph of Page 4. Chimera 1 and chimera 2 were not indicated in Fig.3 We apologize for the careless mistake, which was corrected in the revised manuscript (p. 5, line 3).

Comments: OK

5. LAMP1 was mentioned as an 80 kDa protein in their original paper (Glycobiology vol. 20 no. 8

pp. 976–981, 2010) but was mentioned as a 135 kDa protein in this manuscript. It is better to address this large discrepancy.

We thank the reviewer for the insightful comment. The discrepancy is due to glycosylation differences. To confirm this point, the cell lysate treated with PNGase F was subjected to immunoblotting with the anti-LAMP-1 antibody. The result indicated that the deglycosylated LAMP-1 migrated as a 40-kDa protein, which is consistent with the observation in our previous study. We showed the data in the revised manuscript (p. 3, lines 33-35) by adding Figure 1B.

Comments: OK

Reviewer #3 (Remarks to the Author):

In the revised manuscript, Kato and coworkers have done a fantastic job at addressing all of the reviewers' comments. I find the resulting work of great interest and the data convincing. I therefore recommend the paper for publication and congratulate the authors on this nice piece of work.

Reviewer #1 (Remarks to the Author):

Comments: Based on the above information, we must conclude that the presence of Lewis X epitopes on LAMP-1 inside of cell has no known biological functions. The functions that the author mentioned belong to cell membrane LAMP-1.

As per the reviewer's comments, we modified the text, stating that the intracellular function of Lewis X-carrying LAMP-1 remains unknown (p.5, lines 28-31).

Comments: If desialylated glycoproteins were used for FUT9 modifications, the evidence of desialylation is better to be presented as well. Desialylation usually result in gel mobility shift in SDS-PAGE. Without this evidence, it is hard to make the conclusion that L29 sequence is the major factor for regulating FUT9 modification.

In replying to the reviewer's comment, we presented SDS-PAGE data as the evidence of desialylation (Supplementary Fig. 6).

Besides, Fig 5A is a blot of biotin and anti-Flag, but not anti-Lewis X and anti-Flag as indicated in the figure legend.

We apologize for the careless mistake, which was corrected in the revised manuscript (p.10, lines 3-6 and Fig.5 legend).

Comments: In vitro test on desialylated glycoprotein will be a good experiment. However, there is no evidence of desialylation presented. Desialylation on fetuin will result an obvious gel mobility shift, which can be easily demonstrated, for example. Comparing FUT9 modifications on glycoproteins with and without desialylation is critical to support the claims on L29 sequence, simply because that sialylation can prevent FUT9 modification.

In this manuscript, we conducted the *in vitro* assay using EPO proteins but not fetuin. As mentioned above, we presented the corresponding data for EPO and EPO-L29 as Supplementary Fig. 6.

Response to Reviewer #1

Reviewers' comments:

Reviewer #1 (Remarks to the Editor):

Based on the information provided, I don't know if the authors have done the experiment that I suggested, i.e. provide evidence of desialylation on fetuin that was used in the in vitro fucosylation experiment. This is a very straightforward experiment. Desialylation will cause a band shift in SDS-gel.

As I pointed earlier that sialylation can completely block the fucosylation by FUT9. If they can't provide the sialylation status of fetuin, they can't make a conclusion on the claim of L29 sequence.

Reviewer #1's request was to show evidence that the fetuin used as a substrate was sufficiently desialylated beforehand in the in vitro fucosylation experiment. However, the substrate used in this experiment was NOT fetuin, but EPO, and the data evidence for EPO was shown as Supplementary Fig. 5 in the previous revision.

However, Reviewer#1's request this time was to show evidence of adequate desialylation of fetuin; we do not understand the significance of the request for band shift data on a molecule that was not used as a substrate in the in vitro experiment. Therefore, we did not make any revision in the text.

REVIEWERS' COMMENTS:

Reviewer #1 (Remarks to the Author):

The author indicated in the figure legend of Fig. 5B that desialylated EPO (EPO-L29 should be desialylated as well) was used for the in vitro assay and that they provided evidence of desialylation in Supplementary Fig. 6 accordingly to the recommendation of the reviewer.

Editor's comment:

We therefore invite you to revise your paper one last time to address the remaining concerns of our reviewers. In response to Reviewer #1, please clarify whether the samples in Figure 5b and Supplementary Figure 6 were run on the same blot together and can be directly compared, or not. Please make your response clear in the figure if they were, or include a sentence about this limitation if not.

We confirmed that the samples were fully desialylated based on their band shifts as shown in Supplementary Figure 6. Subsequently the samples were subjected to immunoblotting (Figure 5b). As per the editor's comment, we modified the legends of Figure 5b and Supplementary Figure 6.